# Revealing the Contribution of Informal Settlements to Climate Change Mitigation in Latin America: A Case Study of Isidro Fabela, Mexico City

Ariadna Reyes 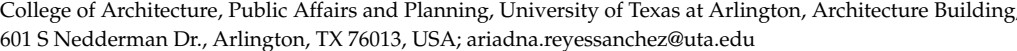

College of Architecture, Public Affairs and Planning, University of Texas at Arlington, Architecture Building, 601 S Nedderman Dr., Arlington, TX 76013, USA; ariadna.reyessanchez@uta.edu

**Abstract:** Given the implications of energy use in housing units for global warming, life cycle assessment (LCA) has been used to examine greenhouse gas (GHG) emissions. Although informal settlements, currently known as colonias populares, or barrios, house most of the urban population in Latin America, there is a poor understanding of how people in these communities use energy and contribute to GHG emissions. This investigation provides a comprehensive analysis of resource consumption in Mexico City's colonia popular, including self-help housing, household energy consumption, and transportation. As there is no spatially specific data on energy use, the author conducted field research in the informal community of Isidro Fabela, founded in the 1960s. Isidro Fabela is an illuminating community that helps understand the energy use of informal settlements at advanced stages of urban consolidation. A mixed-method research approach served to collect empirical data through observations, household surveys, and in-depth interviews. Research sheds light on the efficient and sustainable ways in which families use energy, materials, and resources during self-help construction, and through their daily lives, in their community. The community-based LCA assessment finds that the contribution of Isidro Fabela to GHG emissions is 50% of the average impact in Mexico City. Natural gas or liquefied petroleum gas (LPG) consumption for water heating is the most significant contributor to GHG emissions because families use inefficient heaters. Interestingly, by using public transportation and even walking, residents moderate the contribution of work commutes to GHG emissions. Therefore, climate change policy should enhance low-energy practices in informal settlements, by improving access to energy-efficient technologies and public transportation, to help families contribute further to GHG mitigation in Mexico City and elsewhere in Latin America.

**Keywords:** climate change mitigation; colonias populares; energy use; informal settlements; life cycle assessment; Mexico City; Latin America; sustainability



## 1. Introduction

Nearly one-third of the world's urban population lives in communities that may be considered informal settlements or slums [1]. Many informal communities pose health and safety risks to their residents because of inadequate access to sanitation systems, such as drainage and solid waste removal, as well as city services, such as potable water and electricity. In Latin America, low-income families have improved informal communities over time through their self-help housing efforts [2]. After fifty years of urbanization, many old informal communities in Mexico City have relatively easy access to services, such as electricity and transportation, but they have design deficiencies, such as faulty sewers, and face land tenure issues. Today, nearly two-thirds of the urban population in Mexico City live in a consolidated informal settlement, known as a *colonia popular* or *barrio* [3]. However, there is little understanding of how families in these communities use energy and contribute to greenhouse gas (GHG) emissions.

The contributions of this paper are three-fold. First, this study explores the role of Latin American informal communities in supporting climate change mitigation. Previous research on informal settlements and climate change primarily focused on adaptation strategies to address the sensitivity of precarious infrastructure to extreme climate events [4–6]. There is little understanding of the significance of informal settlements for climate change mitigation [7,8]. Previous research explored the mitigation potential of retrofitting residential buildings, erected by architects and contractors, in cities of the North [9–13]. Few studies on urban metabolism methodological approaches explored the extent of energy consumption in informal communities in the Global South [14–17]. Nevertheless, these studies disregarded the implications of household energy use and transportation energy on greenhouse gas emissions. This investigation contributes to climate change mitigation policy, including Mexico's Nationally Determined Contribution (NDC), which poorly examines the extent of energy use and GHG emissions in low-income communities across Mexican cities. This study informs climate change mitigation research by offering a framework for a more inclusive analysis, of energy use and GHG emissions, associated with families and self-built dwelling units in Mexico City's informal settlements [18,19].

The second contribution is methodological because this research draws on life-cycle assessment (LCA) to thoroughly assess the complex nature of energy use associated with a housing unit's life cycle. Data about a housing unit's life cycle can be separated into two main life cycle phases: embodied and operating energy. Embodied energy is influenced by the characteristics of the building materials employed in the construction of dwelling units. The operating phase includes the use of energy (electricity, natural gas, fuels, and water) in residential buildings and, critically, residential transportation. Transportation energy use is influenced by urban density and the spatial location of buildings within cities, which determines job commute times. Despite the significance of transportation energy in residents' contributions to GHG emissions, transportation is rarely included in the scope of previous LCAs of residential land use [13,20–24]. For instance, previous LCA studies in Mexico focused on formal affordable housing development, led by contractors who follow building codes, thus disregarding informal settlements [18,19]. Using LCA, this study addresses the gaps in energy use theory research by expanding LCA in informal communities.

Third, this research adds resolution to the analysis of energy use and climate change mitigation in Mexico City's informal communities. This study offers a complete examination of GHG emissions, including the process of self-help construction, daily consumption of energy in self-built dwelling units, and peoples' commutes. Because there is no spatially specific data on energy use in Mexico that informs a complete LCA of informal settlements, the investigation required the collection of original data via field research. This study offers community-level data collected through fieldwork in an illuminating colonia popular of Mexico City. The LCA of Isidro Fabela was informed by a mixed-method approach that served to collect energy data. Field research combined qualitative methods used in planning, including walks in the community, observations, and in-depth interviews, with quantitative methods, including energy use and origin-destination surveys conducted between January and July 2017. Surveys served to estimate water, electricity, and liquefied petroleum gas (LPG) consumption; in-depth interviews document the perspectives, stories, and daily practices of energy and resource consumption.

This research also informs energy poverty research in Mexico, which states that social inequalities lead to uneven energy consumption and the inability of many families to increase their energy consumption, partly because of their vulnerable household economies [14,25–27]. More specifically, the LCA of Isidro Fabela revealed that families there consume fewer energy resources and thus contribute less to GHG emissions than wealthier communities in Mexico City. Although this low energy consumption is beneficial for climate change mitigation in Mexico City, it may deteriorate the quality of life because low-income families lack access to technological innovations and thus enact saving practices using rudimentary tools. For example, families take short showers to save energy and water but use inefficient water heaters.

This article proceeds as follows. First, the study reviews past literature on LCAs of residential land use. Then, the paper describes the research methods and examines energy use and GHG emissions in the case study. Finally, drawing on the LCA of Isidro Fabela, the study offers recommendations for supporting GHG mitigation and energy justice in Mexico City's colonias populares.

## 2. Materials and Methods

This section explains the use of LCA as a methodological approach to systematically examine energy use in Mexico City's consolidated informal settlements. A literature review served to select the research methods that helped collect energy data, for conducting a thorough LCA, in a colonia popular in Mexico City. The scope of this LCA includes embodied energy and operating energy. Embodied energy consists of the resource consumption from materials' manufacture and transport, as well as incremental self-help consolidation. This article uses the term "self-help consolidation" to refer to the process in which families enlarge their dwelling units by adding residential extensions and subdividing existing spaces to create separate living spaces and, at the same time, improve the building quality of residential structures. Operating energy consists of household consumption of energy and water, as well as gasoline for work commutes. This study included water consumption because it directly affects the amount of gas used for water heating. Nevertheless, demolition was excluded from the LCA because self-help buildings in Isidro Fabela are being transferred from founders to their adult children over time. Dwelling units in Latin America are rarely demolished because buildings are the central patrimony for low-to-moderate-income families [28].

The unit of analysis is the "housing unit," which helps examine the energy use of self-help buildings and the people who use them. The housing unit (the subject of the LCA) includes three subsystems: (1) dwelling unit, (2) household resource consumption, and (3) work commutes. LCA inventories help to estimate energy consumption in the three subsystems of the housing unit. An LCA inventory integrates the flows of energy and building materials required for the construction and operating phases. First, building materials, such as concrete, require raw materials such as gravel, while transporting materials to construction sites requires gasoline or diesel. This study solely includes the embodied energy associated with building materials for self-help housing, excluding embodied energy related to domestic appliances. Second, household energy use involves flows of electricity, gas for cooking, and water. Additionally, residential transportation requires gasoline for powering automotive vehicles. To ensure the validity of the LCA inventory, the author triangulated empirical data from surveys, with data presented in statistics, developed by federal and local institutions in Mexico.

### 2.1. A Case Study

The literature review revealed gaps in the energy data, published by the Energy Regulator of Mexico (Comisión Reguladora de Energía), which documents the consumption of residential energy use in Mexico. Energy data (such as data on residential electricity consumption) are aggregated at the municipal or state scales, making it difficult to distinguish between the energy use of consolidated informal settlements and that of nearby, upper-income neighborhoods. Because of this lack of fine-grained data on energy and resource consumption, the author conducted field research in Isidro Fabela between January and August 2017.

In the 1960s, impoverished families informally occupied land surrounding the South of Mexico City. They founded Isidro Fabela on a steep slope of rocky soil that completely lacked access to basic services. Using local materials, such as natural stone, and rudimentary construction tools, self-help builders incrementally developed the community infrastructure. For three decades, families here faced precarious living conditions that endangered their health and safety. In the 1980s, the Mexico City government implemented regularization programs in Isidro Fabela that allowed founders to become de-facto owners



of lots. The regularization process led to the gradual provision of basic services, including water and electricity. However, most buildings in Isidro Fabela do not have sewers connected to the municipal drainage; instead, buildings dispose of greywater through fissures in rocky soil, which leads to the deterioration of structures and the exposure of some families to water pollution.

Isidro Fabela is a highly dense community, with a population density of 230 people per hectare, a density 2.5 times greater than that of Mexico City [2]. In Isidro Fabela, families take advantage of relatively large lots of 200 m$^2$ to build dwelling units that house three separate families. The average size of families is four members; thus, self-help buildings in Isidro Fabela house approximately twelve residents.

To document household income in Isidro Fabela, the author asked participants about the characteristics of family workers, including their occupation and education level. This information served to classify formal workers employed by a company and informal workers, such as street vendors and merchants. The average household income in Isidro Fabela may be lower than Mexico City. In Isidro Fabela, workers have lower rates of college education (22%) and higher rates of informal employment (62%) than the average of Mexico City with 34% and 49%, respectively [29,30].

Figure 1 illustrates the spatial location of Isidro Fabela, which, after five decades of urban development in Mexico City, provides excellent access not only to high-capacity transportation systems (shown in black lines) but also to the main beltway of Mexico City, known as "Anillo Periférico" (orange lines). Isidro Fabela was selected as an illuminating community that illustrates the socioeconomic characteristics of people and the building quality of self-built dwelling units in informal settlements at an advanced stage of consolidation, and because of the previous work conducted by Ward [2,31,32], which provided a historical perspective to my analysis of energy use and sustainability.

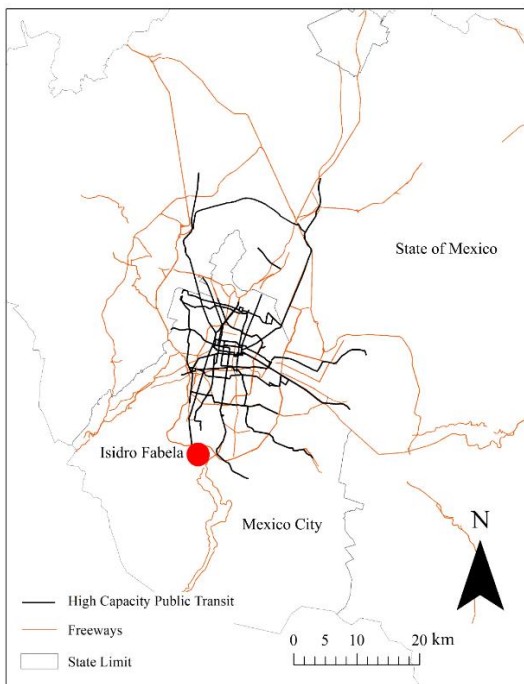

**Figure 1.** The spatial location of Isidro Fabela in the Mexico City Metropolitan Area and its access to transportation systems.

### 2.2. Data Collection Methods

Research methods for estimating energy use include a geospatial analysis, walks, and observations served to establish a sense of the characteristics of self-help buildings, which displayed high heterogeneity in their shapes and sizes. The geospatial analysis served to develop maps of the community that illustrate polygons of parcels. Then, a simple random

sampling of lots served to select the parcels on which architectural observations, energy, and transportation surveys were conducted. Surveys were conducted by knocking on doors and asking heads of households for their consent to participate in the study. Appendix A shows the questions that survey participants responded to during the fieldwork. The average response rate was 80%. The random sampling helped assess estimators of energy use, such as the cost of electricity. In this analysis, every lot in Isidro Fabela had the same probability of being assigned to the investigation. According to the property tax records, there are $N = 1710$ lots in Isidro Fabela. The sample size of this research included $n = 108$ lots in Isidro Fabela, which is large enough to ensure a 95% level of confidence.

### 2.2.1. Embodied Resource Consumption

Architectural observations in $n = 108$ self-help buildings served to document the characteristics of materials, and apparent deficiencies, of buildings. From this sample, $n = 6$ representative self-built dwelling units were selected because they illustrate the full range of housing quality, e.g., the lowest and highest quality buildings. The intensive case study methodology [33] was adapted to document the material characteristics of self-built homes. The author worked with two research assistants: one researcher made general measurements of each residential structure, while a second investigator took notes on the dimensions of dwelling units and drew rough sketches of the buildings. The author also conducted interviews with the heads of households to document the history of self-help improvement.

### 2.2.2. Household Operating Resource Consumption

Surveys ($n = 108$) served to estimate household use of electricity, gas, and water use during the operating phase. For example, to estimate electricity and gas, the heads of households responded with how much they paid for their last electricity, gas, and water bills and the frequency of these payments. Conversion factors transformed the cost of household utilities into Mexican Pesos (MXN), units of kWh for electricity, MJ for natural gas, kg for liquefied petroleum gas (LPG), and liters for water. Additionally, the author documented the types of technology that families use for lighting, cooking, and water heating as well as daily practices of household resource consumption.

### 2.2.3. Transportation Energy Consumption

Origin-destination surveys served to document the characteristics and extent of work commutes from Isidro Fabela. These surveys included questions about the type and number of means of transportation used to commute to job locations. These means of transportation include walking, biking, bus, Bus Rapid Transit (BRT), subway, and private cars. The surveys also included questions to document the time spent on each means of transportation.

## 3. A Comprehensive Assessment of Energy Use in Isidro Fabela

This section addresses the research question: what is the energy consumption associated with families and self-help buildings in Isidro Fabela? To that end, the study offers the LCA inventory of resource consumption associated with embodied and operating energy consumption. This analysis also describes the characteristics of tools, materials, and appliances that families use for construction, and daily use, of buildings as well as the means of transportation for work commutes.

### 3.1. Embodied Energy

The LCA inventory of energy and materials served to estimate the embodied resource consumption (Table 1). Observations of $n = 108$ dwelling units revealed that most self-help builders improve structures with durable materials, such as reinforced concrete for columns. Furthermore, they take advantage of local building materials, such as volcanic stone, to develop foundations. For the walls, self-help builders utilize bricks and concrete

blocks, while some used natural stone. However, the observations revealed that only 42% of the buildings have concrete roofs, while 32% have sheet metal roofs, 22% have asbestos roofs, and 4% use plastic sheeting, most likely due to the lower cost of these unstable roofing materials. Such unstable materials are inadequate for structural safety, and they pose a health hazard associated with the deterioration of asbestos roofs and the release of toxic fibers. Floor slabs and roofs are the most complex and expensive construction systems in self-help housing because these structures need to be made of reinforced concrete to resist the load of upper floors [34].

**Table 1.** Characteristics of the building materials employed in the process of self-help construction in Isidro Fabela, *n* = 108 buildings.

| System | Self-Help Housing in Isidro Fabela |
| --- | --- |
| Foundations | Natural volcanic stone |
| Columns and floor slabs | Reinforced concrete |
| Roofs | Reinforced concrete (42%), sheet metal (32%), and asbestos (22%) |
| Walls | 88% bricks, the rest concrete blocks |

Another shortcoming of self-help housing is that people design structures that are larger than necessary to ensure that the buildings can resist earthquakes and support the load of additional upper floors (Figure 2). For example, in Juan's house, they were 45 cm wide, which is more than double the size of typical columns (20 × 20 cm wide). Juan said, in his interview, "I constructed very thick columns to make sure that my house is strong enough for future incremental housing efforts." When self-help structures are larger than necessary, builders overuse, and sometimes waste, reinforced concrete in the construction process. Structures that are too large significantly increase their embodied energy because they require more reinforced concrete.

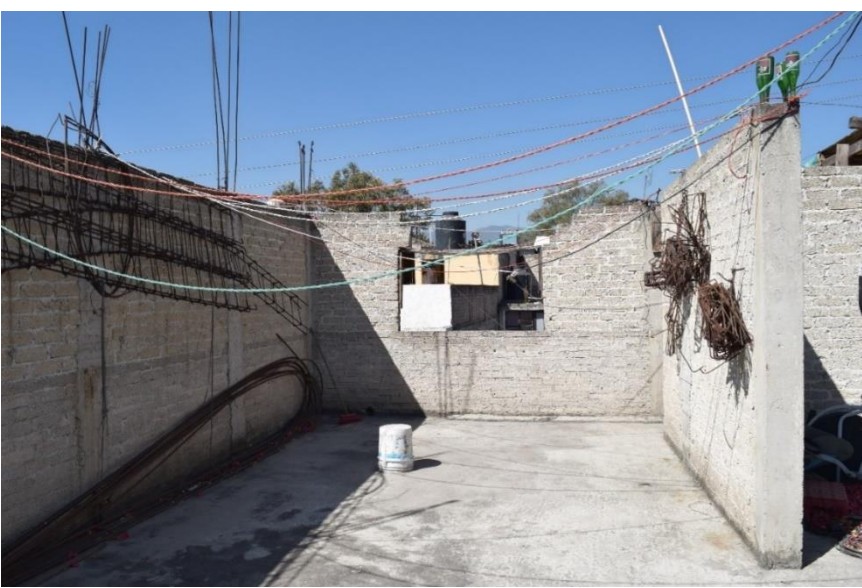

**Figure 2.** The columns of Juan's house.

The fieldwork revealed that self-help builders do not discard materials used in previous construction projects. Instead, they store the materials for reuse in future incremental housing projects. Lucio and Myrna's house represents a typical dwelling unit in Isidro Fabela (Figure 3). Self-help builders reused the asbestos roof, currently on the second floor, after they built a concrete floor slab.

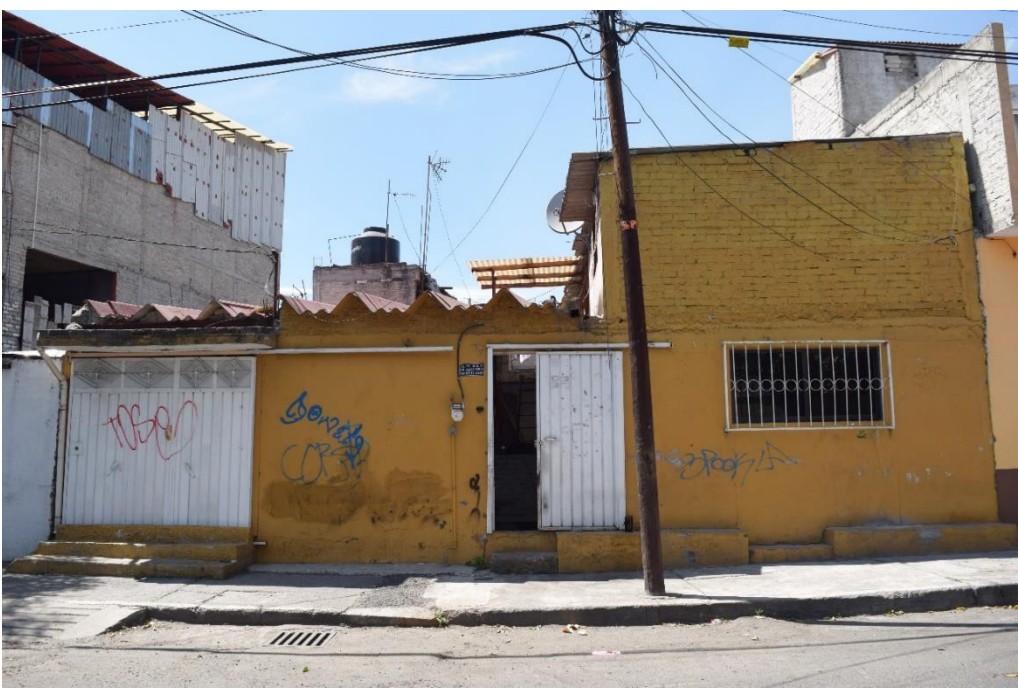

**Figure 3.** Myrna and Lucio's house in 2017.

To facilitate the transportation of materials, self-help builders purchased sacks of cement and steel at "La Casa Blanca," the nearest home improvement store. The distance from Isidro Fabela to the home improvement store was approximately 800 m, and it is a ten-minute walk. Due to the lack of streets and roads in Isidro Fabela, the materials were carried in sacks and, if possible, in wheelbarrows. Self-help builders organized family members and neighbors to move materials, and during construction, people utilize rudimentary construction tools, such as ramps and shovels, to build structures. This indicates that self-help builders completely avoided using diesel or gasoline to move materials to the construction site. "By using our own bodies and labor, rather than machines, such as concrete mixer trucks, self-help builders avoided using fossil fuels to build and improve our homes over time," Ernesto proudly described in his interview.

### 3.2. Operating Phase
Household Energy Consumption

The assessment of household energy use draws on the cost of household utilities, the characteristics of appliances and technologies, and energy consumption practices. This analysis serves to examine the extent to which families use energy and resources in dwelling units. This study compares the resource consumption in Isidro Fabela with the average consumption of families in Mexico City to illustrate energy efficiency.

A cost analysis of household utilities reveals that gas consumption, especially natural gas, is the most expensive household utility, followed by LPG. In Isidro Fabela, 52% of the survey population use natural gas, while the rest acquire cylinders or refill stationary tanks from trucks that sell LPG privately. Electricity is the second most expensive utility in Isidro Fabela, while water service appears to be the most affordable household utility (Figure 4).

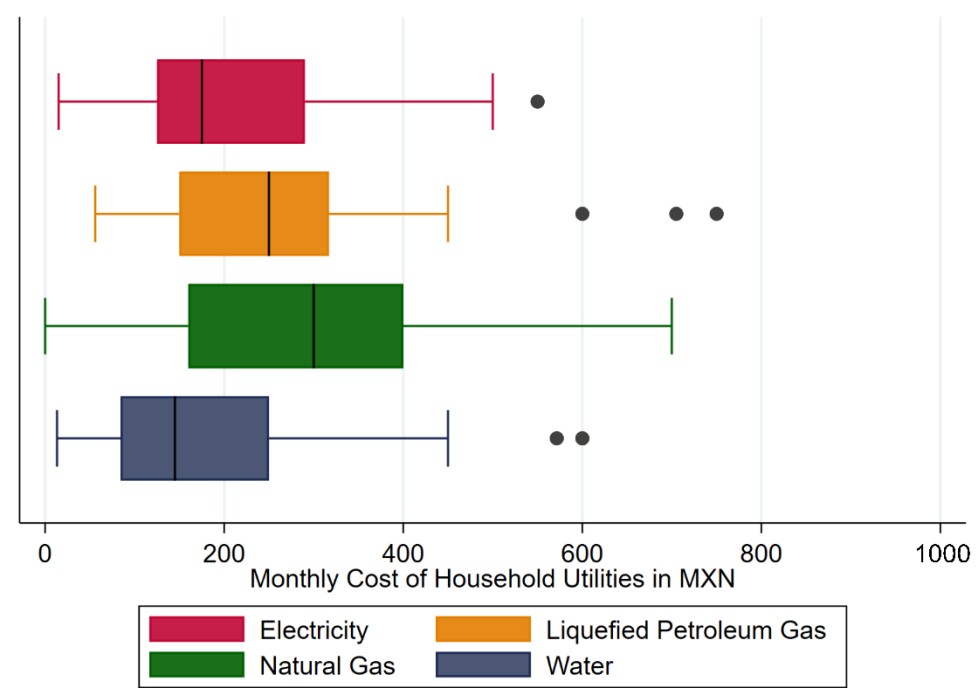

**Figure 4.** Monthly cost of household utilities in Isidro Fabela in Mexican Pesos (MXN).

- Gas

The in-depth interviews revealed that gas, for water heating, is challenging to save because the relatively cold mornings in Mexico City increase the demand for hot water for bathing [35,36]. To save gas, residents take five-minute showers, and some use buckets of hot water that are heated on the stove. One remarkable obstacle for reducing gas usage is residents' lack of access to technological innovations. The survey documented the devices that families use for water heating. Many survey participants (43%) use ineffective cylinder heaters that they operate manually, while 33% of survey respondents have an instant (gas de paso) heater that operates only when needed, rather than continuously. Gas de paso, therefore, helps families save energy, as it shuts off and turns on automatically. Families tend to replace the old cylinder tank with an instant heater to save energy and prevent fire accidents. It was remarkable to discover that 21% of the survey population do not have water heaters. Instead, they use the stove (16%) or inexpensive electric heating devices (5%). Finally, 3% of the survey population reported having a solar water heater. These families expressed pride in having a technological innovation that allows them to use less gas and save money.

- Electricity

Regarding electricity artifacts, the research revealed that 60% of households use fluorescent bulbs, while the rest use incandescent bulbs. While most families use fluorescent bulbs to save electricity, the replacement of incandescent with LED bulbs may further reduce the household consumption of electricity. More importantly, the mild weather of Mexico City throughout the year allows families to avoid using air conditioning systems and heaters to improve indoor temperatures.

The replacement of inefficient refrigerators offers the most significant potential to reduce electricity consumption [37]. Refrigerators that are more than five years old consume significantly more electricity than newer ones that comply with stricter electricity efficiency regulations. In Isidro Fabela, the average age of refrigerators is 9.5 years, which means that most residents have old refrigerators that increase their electricity consumption.

- Water

Water is a fundamental component of household energy consumption and sustainability in Mexico City, which is currently facing a water shortage. In Mexico City, approximately 50% of water is extracted from underground sources, while 30% is pumped from places far from the metropolitan area through pipelines [38]. In Isidro Fabela—and the other consolidated informal settlements in Mexico City—families have time-limited access to potable water from 5 a.m. to 12 p.m.; therefore, residents store water in tanks, buckets, and cisterns or underground tanks [39]. From there, they use electrical pumps to pump the water into a container on the roof, which provides water to the showers, laundry, and other places.

Even though the per-capita cost of water is half that of gas, residents in Isidro Fabela are concerned about water consumption. More specifically, they appear to be aware of Mexico City's water shortage, as Genoveva expressed in her interview, "because we only have a trickle, we have to save water." The survey revealed that all families pursue innovative strategies to save water in their homes; some families pursue more than one water-saving strategy. A third of all families save water and gas by taking short showers, and some residents reuse water from the laundry and bath, while others capture rainwater in buckets during the rainy season.

## 4. The Life Cycle Assessment of Isidro Fabela

This section addresses the research question: how does the contribution of Isidro Fabela families to GHG emissions compare with the average contribution in Mexico City? A comprehensive LCA inventory of the housing unit in Isidro Fabela helps address this question. The LCA inventory includes the embodied and household operating phases.

### 4.1. Life Cycle Inventory

- Embodied energy

To develop the LCA inventory of a self-built housing unit in Isidro Fabela, this study used the measurements of Miguel and Celia's (original founders and residents of Isidro Fabela) dwelling unit. Figure 5 shows the blueprints for the second floor, which resemble the sizes for the first floor. The height of the two-floor dwelling unit is 4.80 m, while the gross floor area is 108 m$^2$.

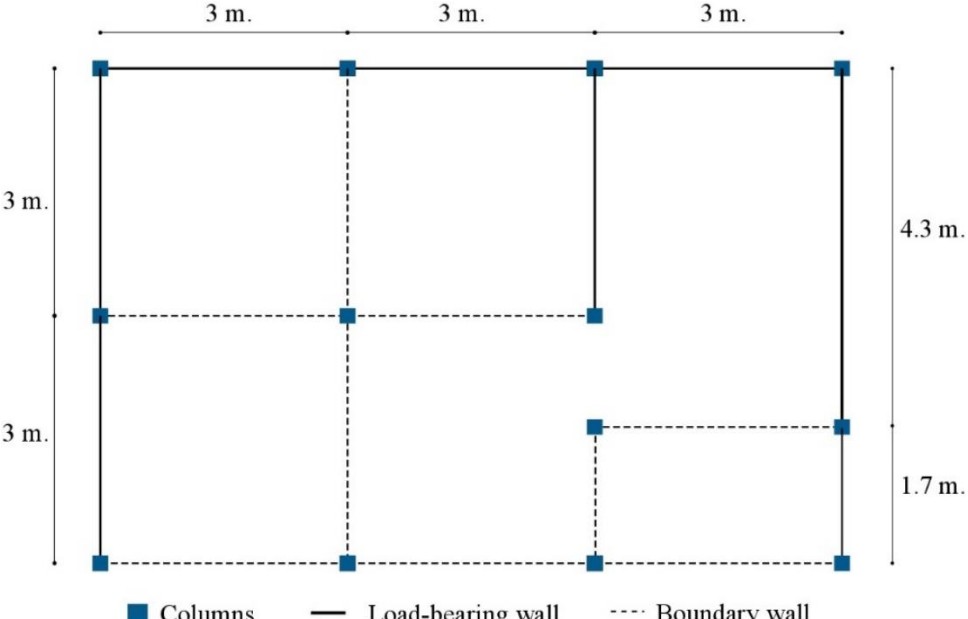

**Figure 5.** Blueprints of Miguel's dwelling unit for the ground floor.

The author created a hypothetical housing unit, built by developers who follow building codes and use heavy machinery and equipment to complete the construction process in a relatively short period, which may take weeks or months. Conversely, the self-help housing construction process in Isidro Fabela is an extremely long process that could take three decades before families have resources to replace discarded materials with durable materials. Despite the shortcomings of self-help housing in Isidro Fabela, dwelling units here are suitable for addressing housing needs in accordance with families' cultural perceptions and, more importantly, accounting for their financial limitations [40].

To compare the embodied energy of dwelling units, this research assumed that buildings have the exact dimensions but differ in the characteristics of materials and finishing. In addition, this research considered structures, such as columns and slabs of 3.6 kPa, which can resist a more significant live load because self-help builders in Isidro Fabela generally develop thicker structures. In contrast, this study considered structures for affordable housing units that resist 2.4 kPa. Another difference is that a housing unit built by developers and contractors usually have envelope materials, such as a layer of asphalt-cellulose, to protect roofs from erosion during the rainy season, which self-built dwelling units in Isidro Fabela do not have.

- Operating energy

The LCA inventory of the operating phase draws on data collected from household surveys. The author used conversion factors to transform the cost of household utilities (in Mexican Pesos) into units of per-capita energy use. The author considered the median cost of household utilities, such as electricity and natural gas, because the distributions had a positive skew.

The author reviewed the electricity tariffs, published by the Federal Electricity Commission in May 2017, which suggested that colonias populares benefit from paying the lowest electricity charges. Likewise, the author reviewed the cost of natural gas and LPG for the same period published by the Electricity and Hydrocarbons Regulator of Mexico (Comisión Reguladora de Energía). Furthermore, this study considers the average cost of water for colonias populares published by the Water Commission in Mexico City.

To examine the extent of efficiency in household consumption, the author compared per-capita energy use in Isidro Fabela with the average consumption in Mexico City (Table 2). The author used figures of average resource consumption in Mexico City published in previous studies. The research revealed that Isidro Fabela residents consume an average of 57% less electricity, 37% less natural gas, and 32% less water than the average consumption in Mexico City. Thus, by performing energy-saving strategies, families in Isidro Fabela use resources efficiently while protecting their household economies.

**Table 2.** Daily per-capita resource consumption in Isidro Fabela and Mexico City. Source: For Mexico City's average consumption of electricity [41], water [42], natural gas [43], and LPG [44].

| Resource | Isidro Fabela | Mexico City |
|----------|---------------|-------------|
| Electricity | 1.24 kWh/day | 2.87 kWh/day for users paying the popular tariff; high-consumption domestic rate is 11.44 kWh/day |
| Natural Gas | 32.4 MJ/day | 51.9 MJ/day (national average) |
| Water | 215 L/day | 314 L/day |

The analysis of transportation energy use in Isidro Fabela included *n* = 214 work commutes, which suggests that the average number of workers per family is 1.98. Origin-destination surveys revealed that 44% of workers stay in the community to work in the local retail economy, such as at grocery stores and street food vending, and thus commute by walking. Although Isidro Fabela is 18 km away from Mexico City's historic center, 41%

of workers commute to job-rich areas in southern Mexico City, mostly in the municipalities of Tlalpan and Coyoacán—roughly 2.5 to 10 km away—while the rest commute to Mexico City's downtown. Origin-destination surveys revealed that the average commute time (one way) in Isidro Fabela is 18.8 min, which is only 35% of the time spent by workers in Mexico City, who commute an average of 54 min, as revealed by the National Institute of Statistics and Geography [45].

As Table 3 explains, 47% of workers commute using various means of public transportation, including low-capacity buses and high-capacity transportation systems, such as the subway and the BRT. Even though 37% of the survey respondents own a private vehicle, only 10% of workers drive a private car for their daily commute to work (about 25 min). This indicates that car ownership does not signify car dependence for daily job commuting. The interviews revealed that car owners do not perceive their cars as their primary mode of transportation; instead, they use their vehicles for social recreation purposes on weekends.

**Table 3.** Comparing transportation practices in Isidro Fabela and Mexico City. Source: For figures of transportation in Mexico City, adapted from INEGI [7].

| Characteristics | Isidro Fabela | Mexico City |
|---|---|---|
| Walking trips | 44% | 26% |
| Public transportation | 46% | 49% |
| Car trips | 10% | 23% |
| Car ownership | 37% | 53.1% |
| Average commute time | 18.8 min (95% confidence interval, 15.9–21.8 min) | 54 min |

Table 4 shows the comprehensive LCA inventory of the housing unit in Isidro Fabela and Mexico City, including embodied and operating energy use. Embodied energy includes materials and energy flows associated with the construction phase. Operating energy draws on the annual consumption of household and transportation energy use associated with four-member households in Isidro Fabela and Mexico City.

**Table 4.** Comprehensive LCA inventory of the housing unit in Isidro Fabela and Mexico City.

| Operating Phase | Flow | Isidro Fabela's Housing Unit | Mexico City's Housing Unit |
|---|---|---|---|
| Embodied energy of building materials | Concrete | 46.7 Tons | 44.2 Tons |
| | Concrete block | 31.7 Tons | 31.7 Tons |
| | Mortar | 30.1 Tons | 30.1 Tons |
| | Aggregate stone | 7.7 Tons | 2.6 Tons |
| | Rebar steel | 5.1 Tons | 3.0 Tons |
| | Natural stone | 4.3 Tons | 0 Tons |
| | Roofing asphalt | 0 Tons | 1.7 Tons |
| | Stucco | 0 Tons | 1.4 Tons |
| Household energy | Electricity | 1814 kWh/year | 4204 kWh/year |
| | Natural gas | 47,389 MJ/year | 75,776 MJ/year |
| Transportation energy | Gasoline | 588 L/year | 1716 L/year |

*4.2. Life-Cycle Assessment and GHG Emissions*

This study used environmental impact assessment methods to evaluate global warming impacts in terms of GHG emissions that result from the estimated flows of energy materials delineated in the LCA inventory. Environmental impact assessment methods transform flows of energy and materials into estimated GHG emissions measured in units of carbon dioxide emissions equivalent ($CO_2$eq). The author used software that applies the tool for reducing and assessing chemical and other environmental impacts (TRACI), which is the LCA method developed by the U.S. Environmental Protection Agency.

The International Energy Agency [46] balance of energy in Mexico's residential sector in 2018 served to provide a panorama of annual energy consumption. The study found that per-capita energy use in Isidro Fabela (0.245 MWh/year) and Mexico City (0.309 MWh/year) represented 44% and 55% of the national estimate for residential buildings (0.558 MWh/year). The temperate weather conditions in Mexico City allow residents to use less energy compared to other Mexican cities with arid and humid weather conditions [47].

This section examines global warming impacts using a functional unit that measures GHG emissions per square meter over a building's life service of 50 years. The LCA found that the total GHG emissions (per m$^2$) in Isidro Fabela of 2725 kg $CO_2$eq represented 50% of the average contribution in Mexico City (5434 kg $CO_2$eq). This study used the International Energy Agency [48] estimate of total $CO_2$eq emissions of an average resident in Mexico (3400 kg $CO_2$eq) to illustrate the extent of the annual carbon footprint. The study found that, per-capita, GHG emissions in Isidro Fabela (2934 kg $CO_2$eq/year) and Mexico City (3400 kg $CO_2$eq/year) represented 43% and 83% of the national estimate for Mexico. This, in turn, means that the average resident in Isidro Fabela contributed 57% less to GHG emissions than the average resident in Mexico in 2018.

Embodied energy of building materials contributed 289 kg $CO_2$eq (per m$^2$) in Isidro Fabela, whereas Mexico City contributed 371 kg $CO_2$eq (per m$^2$). In Isidro Fabela, embodied energy represented 11% of the total GHG emissions during the housing unit's service life, while in Mexico City, construction energy accounted for 7%. These differences can be explained by the fact that the operating phase in Mexico City produces more GHG emissions than in Isidro Fabela by a factor of 2. Overall, operating energy is the major contributor to GHG emissions. For Isidro Fabela, operating energy represented 89% of total GHG emissions, and for Mexico City, it represented 93% (Figure 6). This finding concurs with previous LCA studies in residential land use planning [9,11]. Interestingly, the consumption of natural gas related to water heating is the most significant driver of GHG emissions in Isidro Fabela, with 1356 kg $CO_2$eq (50%) (2169 kg $CO_2$eq [40%] in Mexico City). Therefore, this finding agrees with past GHG assessments in residential buildings [35,49,50] that found that solar water heaters offer the best GHG mitigation potential in Mexico City [18,40,41,50]. The cold mornings in Mexico City increase the demand for hot water, which complicates the ability of families to save natural gas for daily showers. Another obstacle to the efficient consumption of natural gas is the frequent use of inefficient water heaters, which increase gas consumption [37].

Transportation energy is the second-largest contributor to GHG emissions, accounting for 651 kg $CO_2$eq (24%) in Isidro Fabela and 1901 kg $CO_2$eq (35%) in Mexico City. This finding coincides with the LCA assessment of urban housing in Mexico City [51]. Because most workers in Isidro Fabela commute via public transportation and walking, the contribution of transportation energy to GHG emissions is moderate. However, the contribution of transportation energy use may be more significant in informal communities on the fringe that lacks easy access to high-capacity transportation systems, forcing residents to commute extensively by using inefficient means of transportation and cars. This highlights the significance of urban location for sustainable transportation practices [22–24].

Electricity consumption is the smallest contributor to GHG emissions in the operating phase. Household electricity consumption contributed 429 kg $CO_2$eq (16%) in Isidro Fabela and 993 kg $CO_2$eq (18%) in Mexico City. In Isidro Fabela, and elsewhere in Mexico City, families take advantage of the mild weather and perform electricity-saving practices, such as using natural ventilation. The contribution of electricity consumption may be more significant in cities with arid or humid weather conditions, which increase the use of air conditioning systems to improve indoor temperatures.

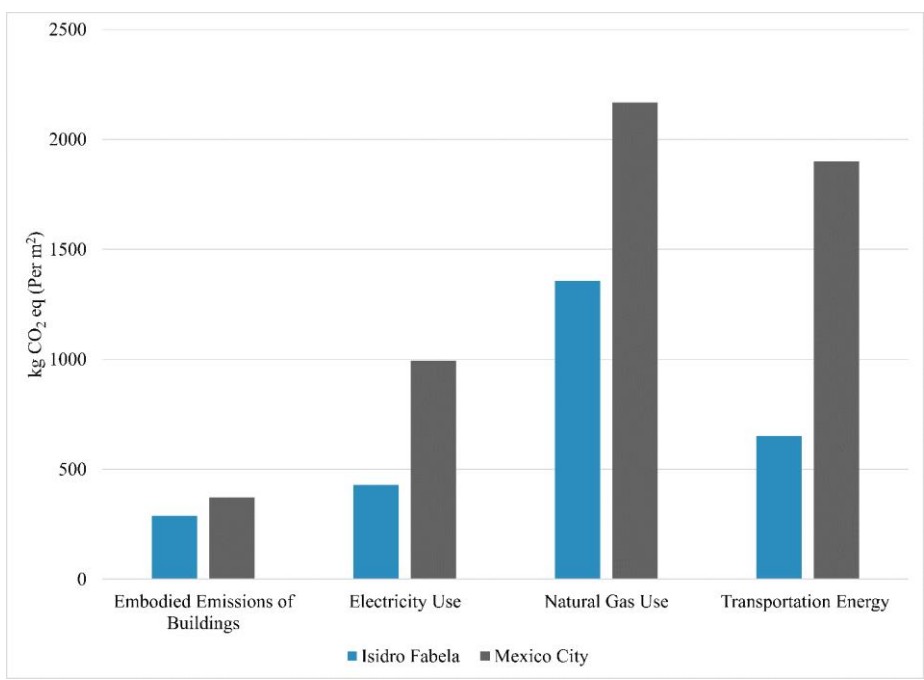

**Figure 6.** Comparison of GHG emissions by life cycle stage and final use of energy in housing units of Isidro Fabela and Mexico City (per m$^2$).

## 5. Conclusions and Policy Implications

In Mexico City, most of the urban land was developed by low-income families through self-help consolidation in informal settlements [3]. However, there is little understanding of how people in informal communities use energy and resources and contribute to GHG emissions. In Mexico, previous research solely examined the extent of energy use of residential buildings in government-funded developments [18,19,51]. This article offers a comprehensive assessment of energy and GHG emissions that combines self-help housing, daily energy consumption, and transportation in the colonia popular of Isidro Fabela.

The complete LCA of Isidro Fabela revealed that families contribute 50% less to GHG emissions than the average household in Mexico City. This analysis reveals disparities in energy use between low-to-moderate income families in Isidro Fabela and wealthier households in Mexico City that can afford to pay more for household utilities. Such inequality in energy use means that upper-income families may be less concerned about the implications of energy use and, consequently, less likely to pursue low-energy practices. For low to moderate-income families in Isidro Fabela, energy use influences household finances. Throughout the housing unit's life cycle, families in Isidro Fabela have likely developed energy-saving behavior to manage household budgets.

Because energy use theory overlooks social justice issues faced by families in informal communities in the Global South, this study draws from the perspective of energy poverty. Energy poverty provides a more inclusive framework to discuss the contribution of informal settlements to climate change mitigation. In doing so, energy poverty helps cities understand the implications of energy use among low- and upper-income populations and thereby mitigate GHG emissions in more socially just ways [25,26,52].

This study reveals that families in Isidro Fabela make efficient use of energy, thus contributing a valuable model for a community that is both energy-efficient and socially just. In Isidro Fabela, three generations of low-to moderate-income families have managed to remain in the community that they developed over the past five decades by using low-energy use practices. Figure 7 illustrates the drivers of energy efficiency derived from the comprehensive LCA in Isidro Fabela.

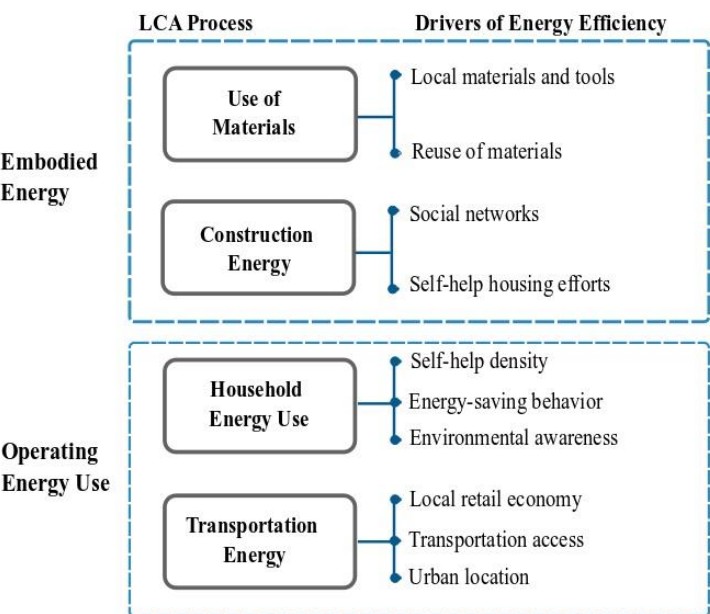

**Figure 7.** Drivers of energy efficiency in the embodied and operating phases in Isidro Fabela.

Despite building relatively large structures that require more reinforced concrete, self-help builders moderate resource consumption by investing their construction labor as well as using and recycling local materials and tools. In addition, social networks of residents in Isidro Fabela enable collective self-help housing efforts to improve community infrastructure and buildings over time.

Regarding the operating phase, the relatively high population density in Isidro Fabela (230 people per hectare) has the side effect of reducing per-capita energy use because up to twelve residents can live in shared dwelling units. People in Isidro Fabela appear to be cognizant of the implications of energy and water consumption in the environmental challenges that Mexico City faces, such as the water shortage and even climate change. In their interviews, many residents expressed interest in acquiring technological innovations, such as solar water heaters, to protect the environment. Previous research found that families may make the best out of technological innovations when they practice energy-saving behaviors [53,54]. The field research suggests that families in Isidro Fabela may take advantage of solar water heaters because natural gas consumption is the most expensive household utility and because they are environmentally aware citizens.

Regarding transportation energy, commuters in Isidro Fabela also moderate fuel consumption by using high- and low-capacity transportation systems or by merely walking, even when some own private cars [55]. The overriding factor that influences efficient work commutes is the urban location and thus the possibility of accessing various means of public transportation and job-rich areas near their homes [51]. The local retail economy is also a significant factor contributing to low transportation energy use, partly because residents work in the community instead of making long commutes [56,57].

This study concurs with previous studies, supporting sustainable upgrading of informal communities in the Global South [5,6,8] to achieve GHG mitigation commitments established in the Paris Agreement on Climate Change (2016) and the 2030 Sustainable Development Goals, including supporting sustainable communities and cities. In addition, Mexico's climate mitigation policy should seek to address energy injustices that exacerbate people's environmental and social vulnerabilities in informal communities. Based on the LCA of Isidro Fabela, this study delineates policy recommendations for supporting climate change mitigation efforts while reducing energy poverty in Mexico.

First, climate change mitigation policy, such as the Nationally Determined Contribution (NDC), should include informal settlements in the scope of its objectives of energy use efficiency [18,19]. The scope of climate change mitigation policy is restricted to assist

government-funded housing developments, disregarding families in informal communities. Therefore, climate policy should improve families' access to technological innovations, including solar water heaters [19,35] and efficient appliances, to mitigate GHG emissions and improve household economies. In this regard, future research on climate change mitigation should test the GHG mitigation potential of technological innovations, as exemplified by previous research in Egypt and Brazil [54,58–60]. In addition, future studies should control the embodied energy implications [61] of replacing domestic appliances with technological innovations.

Second, sustainable housing policies should support assistance to upgrade precarious self-help buildings that endanger the health and safety of families. To that end, technical assistance from construction practitioners and planners may improve the quality of self-help consolidation in informal communities [5,62,63]. Third, as illustrated by Isidro Fabela, the spatial access of consolidated informal settlements to a diverse range of public transportation and the proximity to job-rich areas allow working-poor families to commute efficiently. By supporting the access of working-poor families to public transit, governments can reduce the contribution of transportation energy use to GHG emissions.

Overall, policymakers should enhance the sustainable practices that families in informal settlements already enact by improving low-income families' access to the assistance provided by climate change policies. By revealing the contribution of Isidro Fabela to climate change mitigation in Mexico City, this study may serve as a reference for examining energy use in other informal settlements in the Global South. Mexico City is the second-largest Megacity of Latin America and has higher levels of economic development [64,65] than many of its counterparts in the Global South, but it also has high levels of urban poverty and inequality [66]. Thus, this comprehensive examination of energy and resource consumption, in Mexico City's colonia popular, can serve as a reference for other Latin American cities and beyond in the Global South, which house the largest populations of slums dwellers [7,8].

**Funding:** Research for this article was supported by the School of Architecture and the International Office of the University of Texas at Austin; and the National Council of Science and Technology (CONACYT) in Mexico.

**Institutional Review Board Statement:** The study was conducted according to the guidelines of the Declaration of Helsinki, and approved by the Institutional Review Board of the University of Texas at Austin (IRB Approval Protocol Number 2016-05-0040).

**Informed Consent Statement:** Informed consent was obtained from all subjects involved in the study.

**Data Availability Statement:** The data presented in this study is available on request from the corresponding author. The databases are not publicly available because the confidentiality of participants is protected by the ethics committee of the University of Texas at Austin, TX, United States (IRB Protocol 2016-05-0040).

**Acknowledgments:** The author also wishes to thank Bjorn Sletto and Jacob Wegmann from the University of Texas at Austin, and the anonymous reviewers for their insightful comments.

**Conflicts of Interest:** The author declares no conflict of interest.

### Appendix A. Questionnaire on Operating Energy Use

- Electricity use
    1. How much did the household pay for the last electricity bill?
    2. Does the household have an air conditioning system?
    3. Does the family have any eco-friendly technology, mechanism, or system to save energy, water, or gas at home?
    4. If so, please choose the appropriate answers.
        - Efficient light bulbs
        - Efficient air conditioner

- Solar water heater
- Efficient water heater
- Water-efficient toilet
- Water-efficient shower

- Gas use

  5. What type of gas does the household use for cooking and water heating?

     - Natural gas (pipeline)
     - LP gas (tank)

  6. How much did the family spend on the last gas bill/payment?
  7. What type of water heater does the family have?

     - Traditional (large and cylinder) heater
     - Small, efficient cylinder
     - Electric heater
     - Solar water heater

- Water Use

  8. How much did the household pay for the last water bill?
  9. How much water does the toilet tank use?

     - Big, 11 L
     - Medium, 7 L
     - Small, 4 L
     - Dual mechanism: solids and liquids

  10. Please describe the drainage system in the dwelling unit.

     - Connected to the municipal drainage
     - Septic tanks
     - No drainage system

  11. Does the family have sufficient access to water throughout the day? If not, please indicate the actions that the family members enact to access to water?

     - We built a cistern/tank to store water
     - We recycle water from the laundry
     - We recycle water from the shower
     - We have water-saving appliances, such as efficient showerheads
     - We recycle water
     - We harvest rainwater

- Transportation Energy Use

  12. Where do the family members work?
  13. How do the workers of the family commute to their job locations?

- Which mode of transportation do they usually use, and how long do they spend in every transportation mode?

  14. Does the family own a private car?
  15. Please describe the household's vehicle(s) characteristics and indicate how much drivers spend on gasoline per week?

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
