# Peer review of "Revealing the Contribution of Informal Settlements to Climate Change Mitigation in Latin America: A Case Study of Isidro Fabela, Mexico City"

_sustainability, doi:10.3390/su132112108_

Round 1
Reviewer 1 Report
I thank the authors for making the edits, the quality of the manuscript has been improved. I only have one minor suggestion as follows:
Table 3. The column of "Mexico City" is very hard to read. I suggest reformating this table.
Author Response
- Reviewer 1 states: “I thank the authors for making the edits, the quality of the manuscript has been improved. I only have one minor suggestion as follows:
Table 3. The column of "Mexico City" is very hard to read. I suggest reformating this table.”
Response. Thank you for your kind feedback and careful review. To address this revision, I have revised and simplified the contents of Table 3. Since I deleted Table 1 as suggested by Reviewer 3, this table is currently Table 2.
Table 2. Per-capita resource consumption in Isidro Fabela and Mexico City. Source: For Mexico City’s average consumption of electricity [(CFE, 2018)], water [(Izazola, 2001)], natural gas [(SEDEMA, 2016)], and LPG [(SENER, no date)].
|
Resource |
Isidro Fabela |
Mexico City |
|
Electricity |
1.24 kWh/day |
2.87 kWh/day for users paying the popular tariff; high-consumption domestic rate is 11.44 kWh/day |
|
Natural Gas |
32.4 MJ/day |
51.9 MJ/day (national average) |
|
Water |
215 liters/day |
314 liters/day |

Reviewer 2 Report
Comments
SUMMARY
The paper addresses the research area related to “climate change” of the MDPI Sustainability journal. I believe that the target journal is an appropriate forum for this article. This investigation addresses the gaps in LCA research by expanding the energy use assessment to serve working-poor families in Mexico City’s consolidated informal settlements.
BROAD COMMENT
This is an important paper about climate change mitigation in Mexico. The introduction section is well written with recent references. However, the author failed to link this paper to the National Determined Contribution (NDC) of Mexico, which is at its second generation (2020). The NDC is the document where the country reveals its plans (5 years) to mitigate climate change across sectors. I suggest the author access both NDC documents of Mexico to link them with the results of his/her research. The contribution of the outputs in this manuscript to the NDC of Mexico is missing in the discussion and conclusion.
SPECIFIC COMMENTS
- I suggest the author combine the sections: 1. Introduction and 2. Life cycle energy assessment of residential land use, and summarize them into one section called Introduction.
- The discussion of the results of this study is too superficial. I suggest the author discuss in-depth the results of the study by comparing them to the findings of previous studies and the National Determined Contribution (NDC) of Mexico in the framework of climate change and the Paris Agreement.
Author Response
- Reviewer states: “This is an important paper about climate change mitigation in Mexico. The introduction section is well written with recent references. However, the author failed to link this paper to the National Determined Contribution (NDC) of Mexico, which is at its second generation (2020). The NDC is the document where the country reveals its plans (5 years) to mitigate climate change across sectors. I suggest the author access both NDC documents of Mexico to link them with the results of his/her research. The contribution of the outputs in this manuscript to the NDC of Mexico is missing in the discussion and conclusion.”
Response. In response to Reviewer 's comments, I carefully revised the "Nationally Determined Contributions 2020 Update," published by the Government of Mexico and the Ministry of Environment and Natural Resources (SEDEMA, for its acronym in Spanish) in 2020. This document situates the role of Mexico in the global efforts to mitigate global GHG emissions and delineates strategies to reduce energy use and GHG emissions in residential buildings. However, although these reports recognize the opportunities provided by climate policy to reduce GHG emissions while supporting poverty alleviation, there is no mention of how informal or even low-income communities in Mexican cities will serve to mitigate GHG emissions. One gap in these reports is the lack of understanding of how people in low-income communities across Mexican cities use energy. In addition, the report fails at providing a heterogenous perspective of the energy use of families from the perspective of income and the weather conditions of the city where families reside.
The NDC completely disregards the energy consumption of self-built homes in informal communities or colonias populares in Mexico partly because of the complexity of collecting data via fieldwork and the resolution of energy data in Mexico. Instead, the NDC report uses data and rough assumptions to estimate energy use in government-funded dwelling units in formally developed housing developments, which is multiplied by the total number of housing units in Mexico at the national level. My investigation is a community-level assessment of people's energy use and buildings in a representative colonia popular in Mexico City.
Because the unit of analysis of the GHG emission inventory of the Nationally Determined Contributions Update strategy is national, aggregating whole the dwelling units in Mexico, and my study is an empirical community-level examination of energy, it is impossible to link the results of my study with the national strategy. Nevertheless, although the scale of analysis is different, my study may contribute to the GHG inventories and national climate change mitigation policy in Mexico by providing a careful understanding of how residents in informal communities in Mexico City use energy to construct and use buildings and transportation. To explain the contributions of my research to energy use assessment research in Mexico City and how this study informs climate policy, including the NDC, I added the following sentences in the paper’s introduction.
“This investigation contributes to climate change mitigation policy, including the Nationally Determined Contribution (NDC), which poorly examines the extent of energy use and GHG emissions in low-income communities across Mexican cities. This study informs climate change mitigation research by offering a framework for a more inclusive analysis of energy use and GHG emissions associated with families and self-built dwelling units in Mexico City’s informal settlements [17,18].
The second contribution is methodological because this research draws on life-cycle assessment (LCA) for thoroughly assessing the complex nature of energy use associated with a housing unit’s life cycle. Data about a housing unit’s life cycle can be separated into two main life cycle phases: embodied and operating energy. Embodied energy is influenced by the characteristics of building materials employed in the construction of dwelling units. The operating phase includes the use of energy (electricity, natural gas, fuels, and water) in residential buildings and, critically, residential transportation. Transportation energy use is influenced by urban density and the spatial location of buildings within cities, which determines job commute times. Despite the significance of transportation energy in residents’ contributions to GHG emissions, transportation is rarely included in the scope of previous LCAs of residential land use [22-27]. For instance, previous LCA studies in Mexico focused on formal affordable housing development led by contractors who follow building codes, thus disregarding informal settlements [17,18]. Using LCA, this study addresses the gaps in energy use theory research by expanding LCA in informal communities.” Lines 56-77
- Reviewer 2 states: I suggest the author combine the sections: 1. Introduction and 2. Life cycle energy assessment of residential land use, and summarize them into one section called Introduction.
Response. I thank Reviewer 2 for the recommendations to improve the contents of my manuscript. To address this revision, I combined 1. Introduction and 2. Life cycle energy assessment of residential land use. This section is entitled, “1. Introduction,” I significantly summarized to avoid repetitions, improve clarity and cohesion. This revision helped me reduce the manuscript’s number of words.
- Reviewer states: The discussion of the results of this study is too superficial. I suggest the author discuss in-depth the results of the study by comparing them to the findings of previous studies and the National Determined Contribution (NDC) of Mexico in the framework of climate change and the Paris Agreement.
Response. To address this revision, I added a significant amount of previous LCA studies connected to the investigation of energy use in housing developments in Mexico City and compared the results of these studies with the LCA in Isidro Fabela.
Please refer to the Discussion section 4.3, …” This finding concurs with previous LCA studies in residential land use planning [9,11]. Interestingly, the consumption of natural gas related to water heating is the most significant driver of GHG emissions in Isidro Fabela, with 1,356 kg CO2eq (50%) (2,169 kg CO2eq [40%] in Mexico City). Therefore, this finding agrees with past GHG assessments in residential buildings [40,50,51] that found that solar water heaters offer the best GHG mitigation potential in the case of Mexico City [18,40,41,50]…
Transportation energy is the second-largest contributor to GHG emissions, accounting for 651 kg CO2eq (24%) in Isidro Fabela and 1,901 kg CO2eq (35%) in Mexico City. This finding coincides with the LCA assessment of urban housing in Mexico City [28]. Because most workers in Isidro Fabela commute via public transportation and walking, the contribution of transportation energy to GHG emissions is moderate. However, the contribution of transportation energy use may be more significant in informal communities on the fringe that lacks easy access to high-capacity transportation systems, forcing residents to commute extensively by using inefficient means of transportation and cars. This high-lights the significance of urban location for sustainable transportation practices [24,25].”
Second, as suggested by the Reviewer 2, in the conclusion, I added sentences that discuss how the results of my study inform climate policy reports, including the Nationally Determined Contribution (NDC) of Mexico in the framework of climate change and the Paris Agreement.
“Regarding transportation energy, commuters in Isidro Fabela also moderate fuel consumption by using high- and low-capacity transportation systems or by merely walking, even when some own private cars [29]. The overriding factor that influences efficient work commutes is the urban location and thus the possibility of accessing various means of public transportation and job-rich areas near their homes [28]. The local retail economy is also a significant factor contributing to low transportation energy use, partly because residents work in the community instead of making long commutes [30,31].
This study concurs with previous studies supporting sustainable upgrading of informal communities in the Global South [6, 7, 54] to achieve GHG mitigation commitments established in the Paris Agreement on Climate Change (2016) and the 2030 Sustainable Development Goals, including supporting sustainable communities and cities. In addition, Mexico's climate mitigation policy should seek to address energy injustices that exacerbate people's environmental and social vulnerabilities in informal communities.
Based on the LCA of Isidro Fabela, this study delineates policy recommendations for supporting climate change mitigation efforts while reducing energy poverty in Mexico. First, climate change mitigation policy, such as the Nationally Determined Contribution (NDC), should include informal settlements in the scope of its objectives of energy use efficiency [17,18]. The scope of climate change mitigation policy is restricted to assist government-funded housing developments disregarding families in informal communities. Climate policy should therefore improve the access of families in informal communities to technological innovations, including solar water heaters [18,40] and efficient appliances, to mitigate GHG emissions and improve household economies. In this regard, future research on climate change mitigation should test the GHG mitigation potential of technological innovations, as exemplified by previous research in Egypt and Brazil [56–60]. In addition, future studies should control the embodied energy implications of replacing domestic appliances with technological innovations.
Second, sustainable housing policy should support assistance to upgrade precarious self-help buildings that endanger the health and safety of families. To that end, technical assistance from construction practitioners and planners may improve the quality of self-help consolidation in informal communities [5,61,62]. Third, as illustrated by Isidro Fabela, the spatial access of consolidated informal settlements to a diverse range of public transportation and the proximity to job-rich areas allow working-poor families to commute efficiently. By supporting the access of working-poor families to public transit, governments can reduce the contribution of transportation energy use to GHG emissions.
Overall, policymakers should enhance the sustainable practices that families in informal settlements already enact by improving the access of low-income families to the assistance provided by climate change policy. By revealing the contribution of Isidro Fabela to climate change mitigation in Mexico City, this study may serve as a reference for examining energy use in other informal settlements in the Global South. Mexico City is the second-largest Megacity of Latin America and has relatively higher levels of economic development [63,64] than many of its counterparts in the Global South but high levels of urban poverty and inequality [65]. This comprehensive examination of energy and resource consumption in a Mexico City’s colonia popular can serve as a reference for other Latin American cities and beyond in the Global South [7,8].”

Reviewer 3 Report
Dear Authors,
Thank you for submitting the manuscript. The subject matter is interesting and important. I have the following suggestions:
- May consider revising the title in the following way- "Revealing the Contribution of Informal Settlements to Climate Change Mitigation in Latin America: A Case Study of Isidro Fabela, Mexico City."
- There is no need for Table 1. You can mention these data within the text.
- Please update Figure 1 into a colour one as the grayscale is confusing.
- LCA was supposed to be the main methodology used in this study. But the study is spreading itself thin when transportation was also incorporated. I would suggest just focusing on informal residential energy use rather than transportation.
- The energy use in informal residential areas is usually low which you have also shown. But there is a significant suppressed demand for energy in those settlements, and asking them to reduce GHG emissions is questionable. The LCA based analysis in this study does not make sense.
- The climate change effect may increase the energy demand for cooling, which you noted, but this study does not contribute anything new.
- Please rewrite the manuscript as there is a lot of redundant text, that can be deleted.
Best of luck!
Author Response
1) Reviewer states: “1.May consider revising the title in the following way- "Revealing the Contribution of Informal Settlements to Climate Change Mitigation in Latin America: A Case Study of Isidro Fabela, Mexico City."”
Response. Thank you for your thoughtful revision to improve the title of my manuscript. I have updated the title of my manuscript according to your suggestion.
2) Reviewer states: “There is no need for Table 1. You can mention these data within the text.”
Response. Thank you for your suggestion. To address this revision, I deleted Table 1 and describe the household income data within the text.
To document household income in Isidro Fabela, the author asked participants about the characteristics of family workers, including their occupation and education level. This information served to classify formal workers employed by a company and informal workers, such as street vendors and merchants. The average household income in Isidro Fabela may be lower than Mexico City. In Isidro Fabela, workers have lower rates of college education (22%) and higher rates of informal employment (62%) than the average of Mexico City with 34% and 49%, respectively [34,35].
3) Reviewer states: “Please update Figure 1 into a colour one as the grayscale is confusing.”
Response. Thank you for your careful revision. To address this revision, I reviewed my map and deleted the confusing gray layer that shower urban development growth per decade. I deleted this layer because it is not relevant for the map. Instead, I highlighted the access of the community of Isidro Fabela to high-capacity public transit (black lines) and freeways (yellow lines). Please see the revised map below.
4) Reviewer states: “LCA was supposed to be the main methodology used in this study. But the study is spreading itself thin when transportation was also incorporated. I would suggest just focusing on informal residential energy use rather than transportation.”
Response. Thank you for your revision. One of the main contributions of my research is that it expands the traditional scope of LCA, which solely includes embodied energy and household energy, to include transportation energy. Scholars have highlighted the significance of transportation energy use to understand the implications of urban location, density, and access to transportation for GHG emissions (Norman, MacLean, and Kennedy, 2006; Stephan, Crawford, and de Myttenaere, 2012).
Transportation energy is the most significant contributor to GHG emissions in Megacities of the Global South, including Mexico City (Chavez-Baeza & Sheinbaum-Pardo, 2014) because workers commute extensively from their residences to their job locations (Guerra, 2014). Interestingly, because Isidro Fabela has a strong local retail economy that allows workers to work near their homes, and because of the excellent access to public transit, transportation energy here is moderate and the second-largest contributor to GHG emission. Thus, my research sheds light on the contribution of convenient access of informal communities in urban locations to public transit, as exemplified by Isidro Fabela, to the low use of transportation energy, such as gasoline consumption, for their job commutes.
5) Reviewer states: “The energy use in informal residential areas is usually low which you have also shown. But there is a significant suppressed demand for energy in those settlements, and asking them to reduce GHG emissions is questionable. The LCA based analysis in this study does not make sense.”
Response. Thank you for your revision. I concur with your argument. Energy use in informal settlements, as exemplified by Isidro Fabela, is little. However, this may be detrimental to the quality of life of residents.
Please refer to the introduction:
This research also informs energy poverty research in Mexico, which states that social inequalities lead to uneven energy consumption and the inability of many families to increase their energy consumption partly because of their vulnerable household economies [13,19–21]. More specifically, the LCA of Isidro Fabela revealed that families there consume fewer energy resources and thus contribute less to GHG emissions than wealthier communities in Mexico City. Although this low energy consumption is beneficial for climate change mitigation in Mexico City, it may deteriorate the quality of life because low-income families lack access to technological innovations and thus enact saving practices using rudimentary tools. For example, families take short showers to save energy and water but use inefficient water heaters. (Lines 92-101).
My manuscript is not asking low-income families to use less energy and reduce their contribution to GHG emissions. Instead, my manuscript calls for climate change mitigation policies that help improve families' access to technological innovations. In other words, climate policy should include informal settlements in the scope of its assistance to address energy inequalities, mitigate GHG emissions, and support social justice.
Please refer to the conclusion.
The complete LCA of Isidro Fabela revealed that families contribute 50% less to GHG emissions than the average household in Mexico City. This analysis reveals disparities in energy use between low-to-moderate income families in Isidro Fabela and wealthier households in Mexico City that can afford to pay more for household utilities. Such inequality in energy use means that upper-income families may be less concerned about the implications of energy use and consequently less likely to pursue low-energy practices. For low to moderate-income families in Isidro Fabela, energy use influences household finances. Throughout the housing unit's life cycle, families in Isidro Fabela have likely developed energy-saving behavior to manage household budgets.
…This study concurs with previous studies supporting sustainable upgrading of informal communities in the Global South [6, 7, 54] to achieve GHG mitigation commitments established in the Paris Agreement on Climate Change (2016) and the 2030 Sustainable Development Goals, including supporting sustainable communities and cities. In addition, Mexico's climate mitigation policy should seek to address energy injustices that exacerbate people's environmental and social vulnerabilities in informal communities.
Based on the LCA of Isidro Fabela, this study delineates policy recommendations for supporting climate change mitigation efforts while reducing energy poverty in Mexico. First, climate change mitigation policy, such as the Nationally Determined Contribution (NDC), should include informal settlements in the scope of its objectives of energy use efficiency [17,18]. The scope of climate change mitigation policy is restricted to assist government-funded housing developments disregarding families in informal communities. Climate policy should therefore improve the access of families in informal communities to technological innovations, including solar water heaters [18,40] and efficient appliances, to mitigate GHG emissions and improve household economies. In this regard, future research on climate change mitigation should test the GHG mitigation potential of technological innovations, as exemplified by previous research in Egypt and Brazil [56–60]. In addition, future studies should control the embodied energy implications of replacing domestic appliances with technological innovations.
Second, sustainable housing policy should support assistance to upgrade precarious self-help buildings that endanger the health and safety of families. To that end, technical assistance from construction practitioners and planners may improve the quality of self-help consolidation in informal communities [5,61,62]. Third, as illustrated by Isidro Fabela, the spatial access of consolidated informal settlements to a diverse range of public transportation and the proximity to job-rich areas allow working-poor families to commute efficiently. By supporting the access of working-poor families to public transit, governments can reduce the contribution of transportation energy use to GHG emissions.
Overall, policymakers should enhance the sustainable practices that families in informal settlements already enact by improving the access of low-income families to the assistance provided by climate change policy. By revealing the contribution of Isidro Fabela to climate change mitigation in Mexico City, this study may serve as a reference for examining energy use in other informal settlements in the Global South. Mexico City is the second-largest Megacity of Latin America and has relatively higher levels of economic development [63,64] than many of its counterparts in the Global South but high levels of urban poverty and inequality [65]. This comprehensive examination of energy and resource consumption in a Mexico City’s colonia popular can serve as a reference for other Latin American cities and beyond in the Global South, which house the largest populations of slums dwellers [7,8].”
6) Reviewer states: “The climate change effect may increase the energy demand for cooling, which you noted, but this study does not contribute anything new.”
Response. Thank you for your revision. However, my study does not explore the climate change effect on the demand for cooling. I respectfully argue that this revision is out of the scope of my research. Instead, my study examines how families in an old informal community in Mexico City use energy incrementally to develop self-built dwelling units, use energy and water in their homes, and use fossil fuels to commute to their jobs. My field research found that that families in Isidro Fabela do not use air conditioning systems in their homes because they take advantage of the temperate weather of Mexico City throughout the year. They use natural ventilation and daylighting instead of using mechanical systems, such as air conditioning systems.
7) Reviewer states: Please rewrite the manuscript as there is a lot of redundant text, that can be deleted.”
Response. Thank you for your revision. As suggested by Reviewers 2 and 3, I significantly summarized the content of my manuscript. For instance, I combined the introduction with the literature review. Also, the thoughtful revisions allowed me to reduce the text, avoid repetitions, and improve clarity and cohesion. The length of the manuscript was reduced after the edits.
Again, I thank the three Reviewers for their thoughtful and helpful comments, which have allowed me to revise and improve my article significantly.

Round 2
Reviewer 2 Report
I have undertaken a review of the manuscript (revised) as well as the attached author responses to the initial review where I recommended major revisions. I am satisfied with the revisions made by the authors as they have addressed most, if not all, of my initial comments. Therefore, I do believe that the manuscript has been significantly improved and now warrants publication in Sustainability.

Author Response
I thank Reviewer 2 for thoughtful and helpful comments, which have allowed me to revise and improve my article significantly.
Reviewer 3 Report
Dear Author, Thank you for the revised manuscript.
Author Response
I thank Reviewer 3 for thoughtful and helpful comments throughout the revision process.
This manuscript is a resubmission of an earlier submission. The following is a list of the peer review reports and author responses from that submission.
Round 1
Reviewer 1 Report
I enjoyed reading the article. It has some good contribution specially on several analysis. Here are my specific comments:
- Page 1, line 27: I would define informal settlement as what it really means in Mexco.
- Figure 4, please use colored shaded area to represent different parameters. For example it is very hard to see which one is electricity and which one is LPG. Or write the legends next to the graphs.
- The type of questions asked during teh survey to collect data can be added in an appendix. Also a sample analysis can be included in the appendix that will help other researchers to reproduce.
Reviewer 2 Report
Comments
SUMMARY
The paper addresses the research area related to “climate change” of the MDPI Sustainability journal. I believe that the target journal is an appropriate forum for this article. This investigation, therefore, addresses the gaps in LCA research by expanding the energy use assessment to serve working-poor families in Mexico City’s consolidated informal settlements.
BROAD COMMENT
This is an important paper about climate change mitigation in Mexico. The introduction section is well written with recent references. However, the author failed to link this paper to the National Determined Contribution (NDC) of Mexico, which is at its second generation (2020). The NDC is the document where the country reveals its plans (5 years) to mitigate climate change across sectors. I suggest the author access both NDC documents of Mexico to link them with the results of his/her research. The contribution of the outputs in this manuscript to the NDC of Mexico is missing in the discussion and conclusion.
SPECIFIC COMMENTS
- In the entire manuscript, the citations in the text do not follow the Sustainability journal style. I recommend the author to read the journal's guidelines for authors regarding citations and references.
- I suggest the author combine the sections: 1. Introduction and 2. Life cycle energy assessment of residential land use, and summarize them into one section called Introduction.
- I suggest the author check in the journal’s guideline whether he can use “I” in the entire manuscript. I would recommend using impersonal passive voice form to avoid using “I” throughout the manuscript.
- Lines 324-325: I suggest the author replace the photo of Figure 2 with another where the person does not appear in it. The focus here is the building not the owner of the building.
- Lines 356-358: The unit of the figures on the x-axis is missing on the graph (Figure 4). I suggest the author revise the graph.
- The discussion of the results of this study is too superficial. I suggest the author discuss in-depth the results of the study by comparing them with the findings of previous studies.
Reviewer 3 Report
The study provided a comprehensive life cycle assessment (LCA) with selecting Isidro Fabela as the case study. The manuscript is well-written and with good quality. The methodology is well-designed with detailed information. The findings are original. I believe the methods and conclusions of this paper will benefit the research community in using LCA to examine GHG emissions, especially for the emerging economy.
Below are some minor comments:
Affiliation info is incomplete.
Data Availability Statement: Please refer the journal's requirements on the data availability statement at: https://www.mdpi.com/journal/sustainability/instructions.